# Improving Dynamic Object Interactions in Text-to-Video Generation with AI Feedback

## Abstract

Large text-to-video models hold immense potential for a wide range of downstream applications. However, they struggle to accurately depict dynamic object interactions, often resulting in unrealistic movements and frequent violations of real-world physics. One solution inspired by large language models is to align generated outputs with desired outcomes using external feedback. In this work, we investigate the use of feedback to enhance the quality of object dynamics in text-to-video models. We aim to answer a critical question: what types of feedback, paired with which specific self-improvement algorithms, can most effectively overcome movement misalignment and realistic object interactions? We first point out that offline RL-finetuning algorithms for text-to-video models can be equivalent as derived from a unified probabilistic objective. This perspective highlights that there is no algorithmically dominant method in principle; rather, we should care about the property of reward and data. While human feedback is less scalable, vision-language models could notice the video scenes as humans do. We then propose leveraging vision-language models to provide perceptual feedback specifically tailored to object dynamics in videos. Compared to popular video quality metrics measuring alignment or dynamics, the experiments demonstrate that our approach with binary AI feedback drives the most significant improvements in the quality of interaction scenes in video, as confirmed by AI, human, and quality metric evaluations. Notably, we observe substantial gains when using signals from vision language models, particularly in scenarios involving complex interactions between multiple objects and realistic depictions of objects falling.

## 1 Introduction

Large video generation models pre-trained on internet-scale videos have broad applications such as generating creative video content (Ho et al., 2022a; Hong et al., 2022; Singer et al., 2022; Blattmann et al., 2023b), creating novel games (Bruce et al., 2024), animations (Wang et al., 2019), movies (Zhu et al., 2023), and personalized educational content (Wang et al., 2024), as well as simulating the real-world (Yang et al., 2023b; Brooks et al., 2024) and controlling robots (Du et al., 2024; Ko et al., 2023). Despite these promises, even SoTA text-to-video models today still suffer from hallucination, generating unrealistic objects or movements that violate physics (OpenAI, 2024; Bansal et al., 2024; Yang et al., 2024b), generating static scenes, and ignoring specified movement altogether (Figure 1), which hinders the practical use of text-to-video models.

Expanding both the dataset and the model size has proven effective in reducing undesirable behaviors in large language models (LLMs) (Hoffmann et al., 2022). However, when it comes to video generation, this scaling process is more complex. Creating detailed language labels for training text-to-video models is a labor-intensive task, and the architecture for video generation models has not yet reached a point where it can effectively scale in the same way LLMs have (Yang et al., 2024b). In contrast, one of the most impactful advances in improving LLMs has been the integration of external feedback (Christiano et al., 2017; Ouyang et al., 2022). This raises important questions about what kinds of feedback can be gathered for text-to-video models and how such a feedback can be integrated into training to reduce hallucination and enhance the quality of events in video.

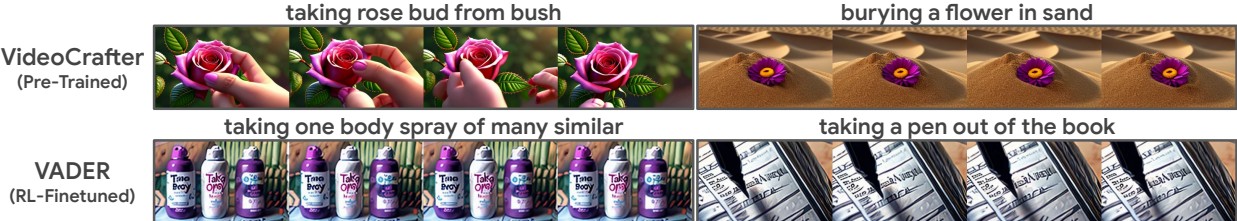

Figure 1: Generated videos from open models: VideoCrafter (Chen et al., 2024) and VADER (Prabhudesai et al., 2024). Both pre-trained and RL-finetuned models fail to represent object movement in the prompts. Improving the dynamics of object interaction in the generated video is still an open problem.

In this paper, we investigate the recipe for improving dynamic interactions with objects in text-to-video diffusion models (Ho et al., 2022b) through external feedback. We cover various self-improvement methods based on reinforcement learning (RL) and automated feedback for text-to-video generation. We begin by noting that two representative offline algorithms, reward-weighted regression (RWR) (Peters & Schaal, 2007) and direct preference optimization (DPO) (Rafailov et al., 2023), introduced independently in prior works, have stemmed from the same objective through the lens of probabilistic inference, with some design choices such as KL-regularization and policy projection. **This equivalence highlights that there is no dominant algorithm in principle; rather, we should care about coupling with reward and data depending on the specific use case.** For feedback choice, we test popular metric-based feedback on alignment (Radford et al., 2021), human preference (Wu et al., 2023a; Kirstain et al., 2023), and dynamics (Huang et al., 2024), and propose leveraging large-scale vision-language models (VLMs) to provide perceptual feedback carefully focusing on object interactions, because VLMs could interpret the video scenes as humans do.

RL-finetuning effectively maximizes various types of reward feedback compared to SFT with self-generated data. Compared to popular video quality metrics in prior works (Prabhudesai et al., 2024; Li et al., 2024b; Yuan et al., 2023), measuring alignment and dynamics, our approach with AI feedback drives the most significant improvements in the quality of interaction scenes; as confirmed by VLMs, human, and quality metric evaluations including smoothness and consistency from VBench (Huang et al., 2024). Qualitatively, pre-trained text-to-video models often struggle with multi-step interactions, new object appearances, and spatial 3D relationships, while RL fine-tuning helps address these limitations. We lastly provide recommended algorithm choices for practitioners.

## 2 Preliminaries

**Denoising Diffusion Probabilistic Models.** We adopt a denoising diffusion probabilistic model (DDPM) (Sohl-Dickstein et al., 2015; Ho et al., 2020) for generating a $H$-frame video $\mathbf{x}_0 = [x_0^1, ..., x_0^H]$. DDPM considers approximating the reverse process for data generation with a parameterized model $p_\theta(\mathbf{x}_0) = \int p_\theta(\mathbf{x}_{0:T}) d\mathbf{x}_{1:T}$, where $p_\theta(\mathbf{x}_{0:T}) = p(\mathbf{x}_T) \prod_{t=1}^{T} p_\theta(\mathbf{x}_{t-1} \mid \mathbf{x}_t)$, and $p(\mathbf{x}_T) = \mathcal{N}(0, I)$, $p_\theta(\mathbf{x}_{t-1} \mid \mathbf{x}_t) = \mathcal{N}(\mu_\theta(\mathbf{x}_t, t), \sigma_t^2 I)$. DDPM also has a forward process, where the Gaussian noise is iteratively added to the data with different noise level $\beta_t$: $q(\mathbf{x}_{1:T}) = \prod_{t=1}^{T} q(\mathbf{x}_t \mid \mathbf{x}_{t-1})$ and $q(\mathbf{x}_t \mid \mathbf{x}_{t-1}) = \mathcal{N}(\sqrt{1 - \beta_t}\mathbf{x}_{t-1}, \beta_t I)$. Considering the upper bound of negative log-likelihood $\mathbb{E}_q[-\log p_\theta(\mathbf{x}_0)]$ with practical approximations, we obtain the objective for DDPM $\mathcal{J}_{\text{DDPM}}$ as follows,

$$\mathbb{E}_q \left[ \sum_{t=1}^{T} D_{\text{KL}}(q(\mathbf{x}_{t-1} \mid \mathbf{x}_t, \mathbf{x}_0) \mid\mid p_\theta(\mathbf{x}_{t-1} \mid \mathbf{x}_t)) \right] \approx \mathbb{E}_{\mathbf{x}_0, \epsilon, t} \left[ \|\epsilon - \epsilon_\theta(\sqrt{\alpha_t}\mathbf{x}_t + \sqrt{\beta_t}\epsilon, t)\|^2 \right] := \mathcal{J}_{\text{DDPM}}, \quad (1)$$

where $\alpha_t = 1 - \beta_t$. After training parameterized denoising model $\epsilon_\theta$, the video might be generated from the initial noise $\mathbf{x}_T \sim \mathcal{N}(\mathbf{0}, I)$ through the iterative denoising from $t = T$ to 1: $\mathbf{x}_{t-1} := \frac{1}{\sqrt{\alpha_t}} \left( \mathbf{x}_t - \frac{\beta_t}{\sqrt{\bar{\beta}_t}} \epsilon_\theta(\mathbf{x}_t, t) \right) + \sigma_t u_t$, where $u_t \sim \mathcal{N}(0, I)$ and $\bar{\beta}_t = \prod_{s=1}^{t} \beta_s$. We also adopt text and first-frame conditioning $\mathbf{c} = [c_{\text{text}}, c_{x_0^1}]$ with classifier-free guidance (Ho & Salimans, 2022), and will consider the conditional formulation $p(\mathbf{x}_0 \mid \mathbf{c})$ in the following sections.

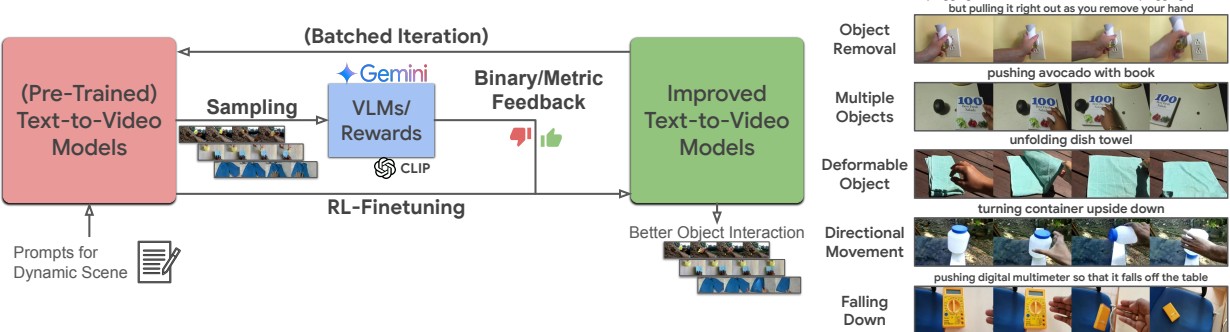

Figure 2: (**Left**) A pipeline for RL-finetuning with feedback. We first generate videos from the pre-trained models, and then use VLMs capable of video understanding (or metric-based reward) to obtain feedback labels. Those data are leveraged for offline and iterative RL-finetuning. We describe the pseudo algorithm in Algorithm 1. (**Right**) Example videos from Something-Something-V2. We define five principles of challenging object movements: object removal (OR), multiple objects (MO), deformable object (DO), directional movement (DM), and falling down (FD).

**Probabilistic Inference View of RL-Finetuning Objectives.** As in Fan et al. (2023) and Black et al. (2024), we formulate the denoising process in DDPMs as a $T$-horizon Markov decision process (MDP), which consists of the state $s_t = (\mathbf{x}_{T-t}, \mathbf{c})$, action $a_t = \mathbf{x}_{T-t-1}$, initial state distribution $P_0(s_0) = (\mathcal{N}(0, I), p(\mathbf{c}))$, transition function $P(s_{t+1} \mid s_t, a_t) = (\delta_{\{a_t\}}, \delta_{\{\mathbf{c}\}})$, reward function $R(s_t, a_t) = r(\mathbf{x}_0, \mathbf{c})\mathbb{1}[t = T]$, and the policy $\pi_\theta(a_t \mid s_t) = p_\theta(\mathbf{x}_{T-t-1} \mid \mathbf{x}_{T-t}, \mathbf{c})$ where $\delta_{\{\cdot\}}$ is the Dirac distribution and $\mathbb{1}[\cdot]$ is a indicator function. Moreover, motivated by the probabilistic inference for control (Levine, 2018), we additionally introduce a binary event variable $\mathcal{O} \in \{0, 1\}$ to this MDP, which represents if the generated video $\mathbf{x}_0$ is optimal or not. To point out the unified RL-finetuning objective for DDPM, we consider the log-likelihood $\log p(\mathcal{O} = 1 \mid \mathbf{c})$ and decompose it with a variational distribution $p'$:

$$\log p(\mathcal{O} = 1 \mid \mathbf{c}) = \mathbb{E}_p \left[ \log p(\mathcal{O} = 1 \mid \mathbf{x}_0, \mathbf{c}) - \log \frac{p(\mathbf{x}_0 \mid \mathbf{c})}{p'(\mathbf{x}_0 \mid \mathbf{c})} + \log \frac{p(\mathbf{x}_0 \mid \mathbf{c})}{p'(\mathbf{x}_0 \mid \mathcal{O} = 1, \mathbf{c})} \right]$$

$$= \mathcal{J}_{\text{RL}}(p, p') + D_{\text{KL}}(p(\mathbf{x}_0 \mid \mathbf{c}) \| p'(\mathbf{x}_0 \mid \mathcal{O} = 1, \mathbf{c})), \tag{2}$$

where $\mathcal{J}_{\text{RL}}(p, p') := \mathbb{E} \left[ \log p(\mathcal{O} = 1 \mid \mathbf{x}_0, \mathbf{c}) - \log \frac{p(\mathbf{x}_0 | \mathbf{c})}{p'(\mathbf{x}_0 | \mathbf{c})} \right]$ is the evidence lower bound. As in the RL literature (Peters et al., 2010; Abdolmaleki et al., 2018; Furuta et al., 2021), we can assume the dependence on the reward such as $p(\mathcal{O} = 1 \mid \mathbf{x}_0, \mathbf{c}) \propto \exp(\beta^{-1} r(\mathbf{x}_0, \mathbf{c}))$, because $\mathcal{O}$ stands for the optimality of the generated video. Then, we could rewrite the unified objective for RL-finetuning in an explicit form:

$$\mathcal{J}_{\text{RL}}(p, p') := \mathbb{E} \left[ \beta^{-1} r(\mathbf{x}_0, \mathbf{c}) - \log \frac{p(\mathbf{x}_0 \mid \mathbf{c})}{p'(\mathbf{x}_0 \mid \mathbf{c})} \right]. \tag{3}$$

The following section will describe that existing algorithms can be derived from this through the policy projection with expectation-maximization ($\mathcal{J}_{\text{f-EM}}$), or with Bradley-Terry assumptions (Bradley & Terry, 1952) ($\mathcal{J}_{\text{r-BT}}$), and characterize both types of objectives in the experiments.

## 3 RL-Finetuning with Feedback

We recover practical algorithms from a unified objective (Section 3.1), provide a brief overview of metric-based feedback for text-to-video models (Section 3.2), and then propose a pipeline for AI feedback from VLMs (Section 3.3).

### 3.1 Connection to Practical Algorithms

Equation 3 represents the general form of RL-finetuning objective for DDPM; which maximizes a reward under the KL-regularization (Levine, 2018). We derive the practical implementation; based on expectation-

maximization, and on the Bradley-Terry assumptions. This equivalence highlights that there is no algorithmic superiority among representative offline methods in principle; rather, we should care about the property of reward and data in the following sections. We cover and compare both algorithmic choices in the following experiments.

**Forward-EM-Projection.** This type of algorithm parameterizes $p' = p_\theta(\mathbf{x}_0 \mid \mathbf{c})$ and $p = p_{\mathrm{ref}}(\mathbf{x}_0 \mid \mathbf{c})$ (i.e., forward KL-regularization), and perform coordinate ascent like a generic EM algorithm; solving $\mathcal{J}_{\mathrm{RL}}$ with respect to the variational distribution $p_{\mathrm{ref}}$ while freezing the parametric posterior $p' = p_{\theta'}$ (E-step), and projecting the new $p_{\mathrm{ref}}^{\mathrm{new}}$ into the model parameter $\theta$ (M-Step). Specifically, E-step converts $\mathcal{J}_{\mathrm{RL}}$ into the constraint optimization problem, and considers the following Lagrangian:

$$
\begin{aligned}
\mathcal{J}_{\mathrm{RL}}(p_{\mathrm{ref}}, \lambda) = &\int p(\mathbf{c}) \int p_{\mathrm{ref}}(\mathbf{x}_0 \mid \mathbf{c})\beta^{-1} r(\mathbf{x}_0, \mathbf{c}) \; d\mathbf{x}_0 d\mathbf{c} \\
&- \int p(\mathbf{c}) \int p_{\mathrm{ref}}(\mathbf{x}_0 \mid \mathbf{c}) \log \frac{p_{\mathrm{ref}}(\mathbf{x}_0 \mid \mathbf{c})}{p_{\theta'}(\mathbf{x}_0 \mid \mathbf{c})} \; d\mathbf{x}_0 d\mathbf{c} + \lambda \left( 1 - \int p(\mathbf{c}) \int p_{\mathrm{ref}}(\mathbf{x}_0 \mid \mathbf{c}) \; d\mathbf{x}_0 d\mathbf{c} \right).
\end{aligned}
\tag{4}
$$

The analytical solution of Equation 4 is $p_{\mathrm{ref}}^{\mathrm{new}}(\mathbf{x}_0 \mid \mathbf{c}) = \frac{1}{Z(\mathbf{c})} p_{\theta'}(\mathbf{x}_0 \mid \mathbf{c}) \exp\left(\beta^{-1} r(\mathbf{x}_0, \mathbf{c})\right)$, where $Z(\mathbf{c})$ is the partition function. M-step projects the non-parametric optimal model $p_{\mathrm{ref}}^{\mathrm{new}}$ to the parametric model by maximizing $\mathcal{J}_{\mathrm{RL}}$ with respect to $p_\theta$:

$$
\mathcal{J}_{\mathrm{RL}}(\theta) = -\mathbb{E}_{p(\mathbf{c})p_{\mathrm{ref}}^{\mathrm{new}}} \left[\log p_\theta(\mathbf{x}_0 \mid \mathbf{c})\right] = -\mathbb{E}_{p(\mathbf{c})p_{\theta'}} \left[ \frac{1}{Z(\mathbf{c})} \exp\left(\beta^{-1} r(\mathbf{x}_0, \mathbf{c})\right) \log p_\theta(\mathbf{x}_0 \mid \mathbf{c}) \right].
\tag{5}
$$

To stabilize the training, RWR (Peters & Schaal, 2007; Lee et al., 2023b) for diffusion models simplifies $\mathcal{J}_{\mathrm{RL}}$ by removing the intractable normalization $Z(\mathbf{c})$, setting pre-trained models $p_{\mathrm{pre}}$ into $p_{\theta'}$, converting exponential transform into the identity mapping, and considering the simplified upper-bound of negative log-likelihood which results in the practical minimization objective:

$$
-\mathbb{E}_{p(\mathbf{c})p_{\theta'}} \left[r(\mathbf{x}_0, \mathbf{c}) \log p_\theta(\mathbf{x}_0 \mid \mathbf{c})\right] \approx \mathbb{E}_{\mathbf{c}, \mathbf{x}_0, \epsilon, t} \left[ r(\mathbf{x}_0, \mathbf{c}) \| \epsilon - \epsilon_\theta(\sqrt{\alpha_t}\mathbf{x}_t + \sqrt{\beta_t}\epsilon, \mathbf{c}, t) \|^2 \right] := \mathcal{J}_{\mathtt{f\text{-}EM}}.
\tag{6}
$$

**Reverse-BT-Projection.** This category parameterizes $p = p_\theta(\mathbf{x}_0 \mid \mathbf{c})$ and $p' = p_{\mathrm{ref}}(\mathbf{x}_0 \mid \mathbf{c})$ (i.e., reverse KL-regularization). For instance, PPO (Schulman et al., 2017; Fan et al., 2023; Black et al., 2024) optimizes $\mathcal{J}_{\mathrm{RL}}$ with a policy gradient. However, such on-policy policy gradient methods require massive computational costs and would be unstable for the text-to-video models. Alternatively, we consider the lightweight approach by optimizing the surrogate objective to extract the non-parametric optimal model into the parametric data distribution $p_\theta$ as in DPO (Rafailov et al., 2023; Wallace et al., 2023). First, we introduce the additional Bradley–Terry assumption (Bradley & Terry, 1952), where, if one video $\mathbf{x}_0^{(1)}$ is more preferable than another $\mathbf{x}_0^{(2)}$ (i.e., $\mathbf{x}_0^{(1)} \succ \mathbf{x}_0^{(2)}$), the preference probability $p(\mathbf{x}_0^{(1)} \succ \mathbf{x}_0^{(2)} \mid \mathbf{c})$ is a function of reward $r(\mathbf{x}_0, \mathbf{c})$ such as $p(\mathbf{x}_0^{(1)} \succ \mathbf{x}_0^{(2)} \mid \mathbf{c}) = \sigma(r(\mathbf{x}_0^{(1)}, \mathbf{c}) - r(\mathbf{x}_0^{(2)}, \mathbf{c}))$, where $\sigma(\cdot)$ is a sigmoid function. Transforming analytical solution of Equation 4 into $p_\theta(\mathbf{x}_0 \mid \mathbf{c}) = \frac{1}{Z(\mathbf{c})} p_{\mathrm{ref}}(\mathbf{x}_0 \mid \mathbf{c}) \exp\left(\beta^{-1} r(\mathbf{x}_0, \mathbf{c})\right)$, we can obtain the parameterized reward $r_\theta(\mathbf{x}_0, \mathbf{c}) = \beta \log \frac{p_\theta(\mathbf{x}_0 \mid \mathbf{c})}{p_{\mathrm{ref}}(\mathbf{x}_0 \mid \mathbf{c})} + \beta \log Z(\mathbf{c})$. The surrogate objective is maximizing the log-likelihood of preference probability by substituting $r_\theta$ into $p(\mathbf{x}_0^{(1)} \succ \mathbf{x}_0^{(2)} \mid \mathbf{c})$ and considering the simplified lower-bound of log-likelihood:

$$
\begin{aligned}
&-\mathbb{E}_{\mathbf{x}_0^{(1:2)}, \mathbf{c}} \left[\log p(\mathbf{x}_0^{(1)} \succ \mathbf{x}_0^{(2)} \mid \mathbf{c})\right] = -\mathbb{E}_{\mathbf{x}_0^{(1:2)}, \mathbf{c}} \left[\log \sigma \left( \beta \log \frac{p_\theta(\mathbf{x}_0^{(1)} \mid \mathbf{c}) p_{\mathrm{ref}}(\mathbf{x}_0^{(2)} \mid \mathbf{c})}{p_{\mathrm{ref}}(\mathbf{x}_0^{(1)} \mid \mathbf{c}) p_\theta(\mathbf{x}_0^{(2)} \mid \mathbf{c})} \right) \right] \\
&\approx -\mathbb{E}_{\mathbf{x}_0^{(1:2)}, \mathbf{c}, \epsilon, t} \left[\log \sigma \left(-\beta(\|\epsilon - \epsilon_\theta^{(1)}\|^2 - \|\epsilon - \epsilon_{\mathrm{ref}}^{(1)}\|^2 - \|\epsilon - \epsilon_\theta^{(2)}\|^2 + \|\epsilon - \epsilon_{\mathrm{ref}}^{(2)}\|^2)\right)\right] := \mathcal{J}_{\mathtt{r\text{-}BT}},
\end{aligned}
\tag{7}
$$

where we shorten $\epsilon_\theta(\sqrt{\alpha_t}\mathbf{x}_t^{(1)} + \sqrt{\beta_t}\epsilon, \mathbf{c}, t)$ as $\epsilon_\theta^{(1)}$ for simplicity. In contrast to $\mathcal{J}_{\mathtt{f\text{-}EM}}$, $\mathcal{J}_{\mathtt{r\text{-}BT}}$ has a negative gradient which pushes down the likelihood of undesirable samples.

## 3.2 Metric-based Reward for RL-Finetuning

With the algorithms described in Section 3.1, we may choose any reward to be optimized. Inspired by prior works on finetuning text-to-image models (Fan et al., 2023; Black et al., 2024), RL-finetuning for text-to-

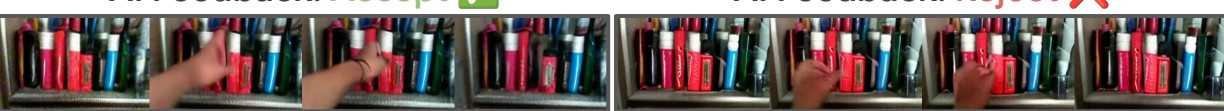

**Prompt: taking one body spray of many similar**

**AI Feedback: Accept** ✅ | **AI Feedback: Reject** ❌

The video shows a shelf with arranged bottles, likely for personal care. A hand enters the frame and picks up a red bottle. The hand then removes the bottle, leaving a gap in the arrangement. The rest of the items remain in place, and the video ends with the shelf slightly rearranged due to the removal.

The video shows a shelf with colorful bottles and containers that are neatly arranged. A hand enters the frame and tries to pick up a red container, but it fails to grasp it properly. The video ends with the shelf as is.

Figure 3: Example of AI feedback and the scene descriptions. We test the capability of understanding physics in VLMs by asking Gemini to evaluate the video of "*taking one body spray of many similar*". VLMs are capable enough to be the best proxy for humans as VLMs capture the events in scene correctly.

video models (Prabhudesai et al., 2024; Li et al., 2024b; Yuan et al., 2023) has also employed metric-based feedback per frame as a reward to enhance the visual quality, style of images, text-content alignment, and optical flow, while summing the scores over the frames. The metric-based rewards are popular because they can amortize the costs to collect actual human feedback (Lee et al., 2023b), and can propagate the gradient through the parameterized evaluators (Clark et al., 2024). Based on previous works, we compare the following diverse metric-based feedback:

**CLIP score** (Radford et al., 2021). Leveraging CLIP is one of the most popular methods to evaluate generative models in text-image alignment, which reflects the semantics of the scene to a single metric. We use ViT-B16 (Dosovitskiy et al., 2020) as an architecture.

**HPSv2** (Wu et al., 2023b;a) **and PickScore** (Kirstain et al., 2023). They could predict a human preference for generated images, because they are trained on large-scale preference data from DiffusionDB (Wang et al., 2022), COCO Captions (Chen et al., 2015), and various text-to-image models. They use OpenCLIP (Ilharco et al., 2021) with ViT-H14 and take both image and text as inputs.

**Optical Flow** (Huang et al., 2024). While assessing semantics per frame, we should also consider the dynamic degree in the generations, which can be captured with the optical flow between the successive frames. Following a prior work (Ju et al., 2024), we use RAFT (Teed & Deng, 2020) to estimate the optical flow, taking the average of norm among the top-5% as a single scalar.

In practice, we adopt reward shaping via linear transformation for each metric-based reward to control the scale of gradients (Fan et al., 2023): $r_{\text{lin}}(\mathbf{x}_0, \mathbf{c}) = \eta r(\mathbf{x}_0, \mathbf{c}) + \gamma$ where $\eta > 0$ is a scaling and $\gamma \geq 0$ is a shifting factor. The scaling factor is important to the optical flow, and the shifting factor to other metrics (CLIP, HPSv2, and PickScore). We test a range of constant factors, such as $\eta \in \{0.005, 0.0075, 0.01, 0.1, 1.0\}$ and $\gamma \in \{0, 0.5, 0.75, 1.0\}$, and report the best results.

### 3.3 AI Feedback from Vision-Language Models

One of the most reliable evaluations of any generative model can be a feedback from humans, while human evaluation requires a lot of costs. One scalable way to replace subjective human evaluation is AI feedback (Bai et al., 2022; Wu et al., 2024c), which has succeeded in improving LLMs (Lee et al., 2023a). Inspired by this, we propose employing VLMs, capable of video understanding, to obtain the AI feedback for text-to-video models.

We provide the textual prompt and video as inputs and ask VLMs to evaluate the input video in terms of overall coherence, physical accuracy, task completion, and the existence of inconsistencies (see Appendix D for the prompts to elicit feedback on the video quality). The feedback VLMs predict is a binary label; accepting

| Methods | CLIP | | HPSv2 | | PickScore | | Optical Flow | | AI Feedback | |
|---|---|---|---|---|---|---|---|---|---|---|
| | Train | Test | Train | Test | Train | Test | Train | Test | Train | Test |
| **Pre-Trained** | 1.9240 | 1.9275 | 1.8139 | 1.7772 | 1.5697 | 1.5637 | 370.2 | 409.6 | 53.02% | 51.56% |
| **SFT** | 1.9287 | 1.9270 | 1.8114 | 1.7748 | 1.5698 | 1.5639 | 368.6 | 416.4 | 55.94% | 47.50% |
| **RWR** ($\mathcal{J}_{\text{f-EM}}$) | **1.9309** | 1.9289 | 1.8132 | 1.7776 | 1.5699 | 1.5644 | 375.0 | **422.0** | **58.23%** | 50.94% |
| **DPO** ($\mathcal{J}_{\text{r-BT}}$) | 1.9259 | **1.9325** | **1.8160** | **1.7797** | **1.5701** | **1.5641** | **377.8** | 410.7 | 56.56% | **55.00%** |

Table 1: Performance of text-to-video models after RL-finetuning, optimizing each reward by each algorithm independently. The color indicates the metric is improved or worsen compared to pre-trained models. RL-finetuning (RWR and DPO) improves most metrics compared to pre-trained models or SFT (no reward signal), and exhibits better generalization to unseen test prompts.

| Algorithm×Reward | Gemini-1.5-Pro Eval | | | GPT-4o Eval | | | Human Eval | | | Overall Consistency (V) | | |
|---|---|---|---|---|---|---|---|---|---|---|---|---|
| | Train | Test | All | Train | Test | All | Train | Test | All | Train | Test | All |
| **Pre-Trained** | 53.02% | 51.56% | 52.66% | 43.26% | 48.05% | 44.45% | 19.79% | 18.13% | 19.38% | 0.1569 | 0.1642 | 0.1587 |
| **SFT** | 55.94% | 47.50% | 53.83% | 45.63% | 44.92% | 45.45% | 23.65% | 13.44% | 21.09% | 0.1580 | 0.1625 | 0.1592 |
| **RWR-CLIP** | 55.31% | 45.00% | 52.73% | 46.02% | 41.25% | 44.82% | 27.19% | 12.50% | 23.52% | 0.1581 | 0.1612 | 0.1589 |
| **RWR-HPSv2** | 52.92% | **57.50%** | 54.06% | 44.04% | **47.81%** | 44.98% | 26.04% | 21.56% | 24.92% | 0.1579 | 0.1636 | 0.1593 |
| **RWR-PS** | 55.52% | 49.69% | 54.06% | 47.37% | 40.70% | 45.70% | 25.73% | 11.56% | 22.19% | 0.1583 | 0.1608 | 0.1589 |
| **RWR-OptFlow** | 57.81% | 50.00% | 55.86% | 47.37% | 37.19% | 44.82% | 28.75% | 10.00% | 24.06% | 0.1590 | 0.1598 | 0.1592 |
| **RWR-AIF** | **58.23%** | 50.94% | **56.41%** | 47.40% | 45.00% | **46.80%** | **33.65%** | **23.44%** | **31.09%** | 0.1594 | 0.1656 | 0.1610 |
| **DPO-CLIP** | 52.29% | 54.38% | 52.81% | 44.58% | 49.92% | 45.92% | 24.90% | 23.44% | 24.53% | 0.1574 | 0.1671 | 0.1598 |
| **DPO-HPSv2** | 55.73% | 51.56% | 54.69% | 42.86% | **51.02%** | 44.90% | 26.35% | 25.00% | 26.02% | 0.1575 | 0.1669 | 0.1599 |
| **DPO-PS** | 53.02% | **55.31%** | 53.59% | 43.62% | 49.06% | 44.98% | 25.94% | 22.50% | 25.08% | 0.1564 | 0.1659 | 0.1588 |
| **DPO-OptFlow** | 54.06% | 54.06% | 54.06% | 44.22% | 48.52% | 45.29% | 26.88% | 24.06% | 26.17% | 0.1567 | 0.1655 | 0.1589 |
| **DPO-AIF** | **56.56%** | 55.00% | **56.17%** | 44.82% | 50.39% | **46.21%** | **36.04%** | **28.13%** | **34.06%** | 0.1577 | 0.1676 | 0.1602 |
| | Subject Consistency (V) | | | Background Consistency (V) | | | Motion Smoothness (V) | | | Imaging Quality (V) | | |
| **PT** | 0.7751 | 0.7816 | 0.7768 | 0.8905 | 0.8910 | 0.8906 | 0.9266 | 0.9262 | 0.9265 | 40.81 | 42.21 | 41.16 |
| **SFT** | 0.7758 | 0.7942 | 0.7804 | 0.8919 | 0.8905 | 0.8915 | 0.9274 | 0.9262 | 0.9271 | 39.93 | 40.79 | 40.15 |
| **RWR-CLIP** | 0.7812 | 0.7828 | 0.7816 | 0.8912 | 0.8941 | 0.8919 | 0.9279 | 0.9261 | 0.9274 | 40.02 | 40.78 | 40.21 |
| **RWR-HPS** | 0.7760 | 0.7930 | 0.7803 | 0.8914 | 0.8961 | 0.8925 | 0.9274 | 0.9281 | 0.9276 | 40.12 | **41.80** | 40.54 |
| **RWR-PS** | 0.7818 | 0.7786 | 0.7810 | 0.8925 | 0.8891 | 0.8917 | 0.9279 | 0.9252 | 0.9273 | 40.14 | 40.60 | 40.26 |
| **RWR-OF** | 0.7853 | 0.7715 | 0.7818 | 0.8934 | 0.8859 | 0.8915 | 0.9292 | 0.9259 | 0.9283 | 40.22 | 39.55 | 40.06 |
| **RWR-AIF** | **0.7855** | 0.7976 | **0.7885** | 0.8936 | 0.8949 | **0.8940** | 0.9299 | 0.9310 | 0.9302 | **40.41** | 41.62 | **40.72** |
| **DPO-CLIP** | **0.7779** | 0.7901 | 0.7809 | 0.8909 | 0.8947 | 0.8918 | 0.9275 | 0.9263 | 0.9272 | **40.24** | 41.88 | **40.65** |
| **DPO-HPS** | 0.7783 | 0.7907 | 0.7814 | 0.8911 | 0.8936 | 0.8917 | 0.9276 | 0.9264 | 0.9273 | 40.15 | 41.76 | 40.55 |
| **DPO-PS** | 0.7769 | 0.7948 | 0.7814 | 0.8910 | **0.8949** | 0.8920 | 0.9273 | 0.9275 | 0.9273 | 40.04 | 41.86 | 40.50 |
| **DPO-OF** | 0.7766 | 0.7908 | 0.7801 | 0.8915 | 0.8944 | 0.8922 | 0.9273 | 0.9276 | 0.9274 | 39.95 | 41.83 | 40.42 |
| **DPO-AIF** | 0.7777 | **0.7956** | **0.7822** | 0.8919 | 0.8949 | **0.8926** | 0.9277 | 0.9285 | 0.9279 | 40.18 | **41.90** | 40.61 |

Table 2: VLMs (Gemini/GPT), human preference, and VBench (V) (Huang et al., 2024) evaluations among the combination of algorithms and rewards. `Algorithm-Reward` stands for finetuning text-to-video models by optimizing `Reward` with `Algorithm`. Compared to other metric rewards popular in video domain, AI feedback from Gemini (RWR-AIF and DPO-AIF) achieves the best quality assessed by Gemini, GPT, and human feedback as well as many VBench scores focusing on consistency and smoothness. As for RWR and DPO, RWR may achieve better alignment on the train split than DPO, while exhibiting *over-fitting* behaviors, where the performance against the test split degrades from the pre-trained models.

the video if it is coherent and the task is completed correctly, or rejecting it if it does not satisfy any evaluation criteria. We mainly use Gemini-1.5-Pro (Gemini Team, 2023) for data generation and evaluation, and also use GPT-4o (OpenAI, 2023) to test the generalization.

To test the physics understanding in VLMs, we conduct preliminary evaluations; we ask Gemini to assess generated videos, and then analyze the feedback and rationale in the response. VLMs score each generation individually (i.e., point-wise), similar to the reward used to train LLMs with human feedback (Qin et al., 2023). As in Figure 3, VLMs could recognize the scene correctly, such as the success or failure of grasping a bottle. In addition, with the a priori that the true video is always preferable, we prepare pairs of the true and generated video, and measure the accuracy of AI feedback. We observe that VLMs can classify the true

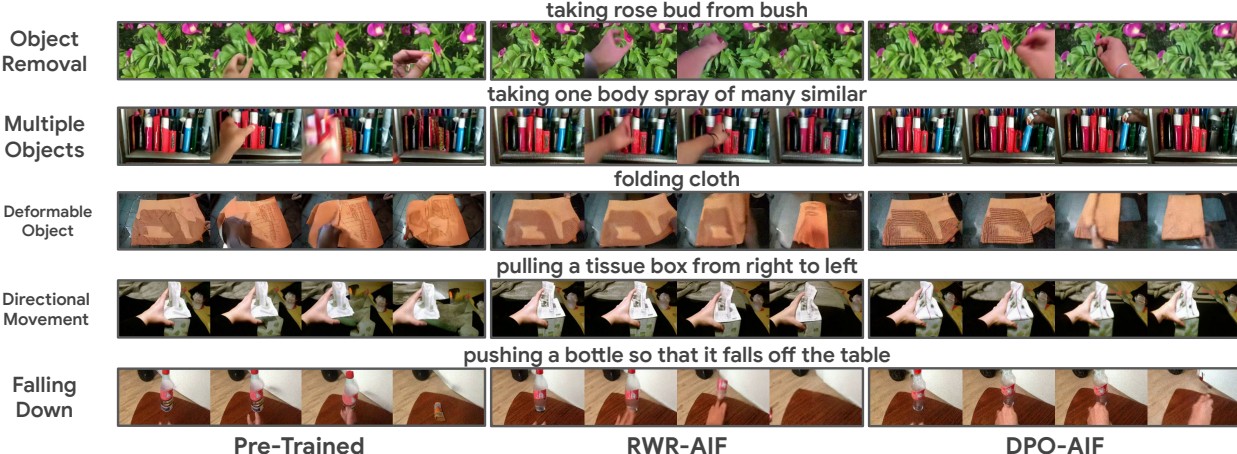

Figure 4: Generated videos from pre-trained models, RWR and DPO. RL-finetuning enhances the quality of dynamic events, where the contents align the prompt and common sense. See Appendix H for further examples.

videos preferable to generated videos for 90.3% of the time, which supports that VLMs are capable enough to simulate human supervision.

## 4 Experiments

We first describe the experimental setup and dataset composition for realistic video generation with dynamic object movement (Section 4.1). In the experiments, we test whether RL-finetuning increases each metric compared to SFT on self-generated data (Section 4.2), and investigate which combination of algorithms and rewards can improve dynamic scene generations the most through VLMs, humans, and VBench (Huang et al., 2024) evaluations (Section 4.3). We also analyze the relationship between human preference and other feedback (Section 4.4), and the performance gap between pre-trained and RL-finetuned models (Appendix G). Lastly, we provide takeaways of algorithm choices for practitioners (Section 4.5).

**Experimental Setup.** We pre-train video diffusion models with 3D-UNet (Özgün Çiçek et al., 2016) (3B parameters in total) with Something-Something-V2 dataset (160K samples) until converged, which obtains a good prior model for realistic object movements. As explained in Section 4.1, we prepare 5×24 prompts for training and 5×8 prompts for evaluation and generate 128 samples per each prompt in the train split from pre-trained models to collect the data for finetuning. We then put AI feedback and reward labels on the generated videos. Figure 2 summarizes the whole pipeline. See Appendix A for the details of model training. For the evaluation, we generate 32 videos per prompt, compute each metric per video summing them over the frames, and average over top-8 samples. Under the same instruction as VLMs for the binary AI evaluation, we also conduct a human evaluation with binary rating.

### 4.1 A Set of Challenging Object Movements

While state-of-the-art text-to-video models can generate seemingly realistic videos, the generated sample often contains static scenes ignoring the specified behaviors, movement violating the physical laws and appearing out-of-place objects (Figure 1), which can be attributed to misalignment, insufficient instruction following, and the lack of physical understanding in video generation (Bansal et al., 2024; Liu et al., 2024b). To characterize the hallucinated behavior of text-to-video models in a dynamic scene, we curate dynamic and challenging object movements from a pair of prompt and reference videos. From empirical observations, we define the following five categories as challenging object movements:

- **Object Removal**: To move something out from the container in the scene, or the camera frame itself. A transition to a new scene often induces out-of-place objects. For example, *taking a pen out of the box.*

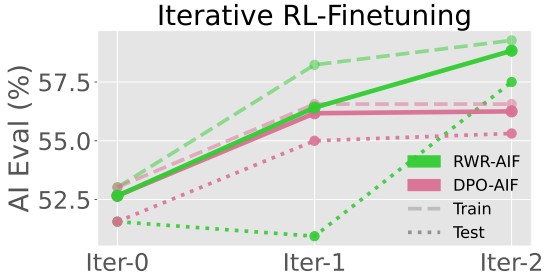

| | Gemini-1.5-Pro Eval | | |
|---|---|---|---|
| Method | Train | Test | All |
| **PT** | 53.02% | 51.56% | 52.66% |
| **DPO-AIF** | 56.56% | 55.00% | 56.17% |
| **VideoCrafter** | 4.17% | 4.17% | 4.17% |
| **CogVideoX-2B** | 11.67% | 8.33% | 10.83% |
| **Wan 2.1-1.3B** | 12.50% | 5.83% | 10.83% |

Figure 5: (**Left**) Preference evaluation after iterative RL-finetuning with AI feedback. RWR-AIF continually improves its outputs by leveraging on-policy samples, while DPO-AIF seems to be saturated. (**Right**) Gemini-1.5-Pro evaluation among recent open models. Open models for text-to-video generation do not have sufficient prior knowledge to express dynamic object interactions. See Appendix F for qualitative results.

- **Multiple Objects**: Object interaction with multiple instances. In a dynamic scene, it is challenging to keep the consistency of all the contents. For example, *moving lego away from mouse.*

- **Deformable Object**: To manipulate deformable objects, such as cloth and paper. The realistic movement of non-rigid instances requires sufficient expressibility and can test physical understanding. For example, *twisting shirt wet until water comes out.*

- **Directional Movement**: To move something in a specified direction. Text-to-video models can roughly follow the directions in the prompts, although they often fail to generate consistent objects in the scene. For example, *pulling water bottle from left to right.*

- **Falling Down**: To fall something down. This often requires the dynamic expression towards the depth dimension. For example, *putting marker pen onto plastic bottle so it falls off the table.*

As a realistic object movement database, we use Something-Something-V2 dataset (Goyal et al., 2017; Mahdisoltani et al., 2018). We carefully select 32 prompts and videos per category from the validation split, using 24 prompts for train data and 8 for test data (Figure 2). See Appendix C for the list of prompts. Note that these prompt sets (train: 120, test: 40 prompts) for RL-finetuning are significantly larger and more diverse than prior works. For instance, Black et al. (2024) use 3 and Fan et al. (2023) use 4 prompts while finetuning one model per prompt. In contrast, we train a single model with hundreds of prompts at once to improve generalization. For the evaluation, we test whether generated samples can be seen as realistic through the human and VLM evaluation as well as VBench (Huang et al., 2024) – a standard protocol to measure the perceptual video quality (Kim et al., 2024a; Oshima et al., 2025).

## 4.2 RL-Finetuning Works Better than SFT

Table 1 presents the video quality, measured by each reward, in text-to-video generation after finetuning. We finetune and evaluate the models for each combination of feedback and algorithm independently. RWR and DPO, RL-finetuning with feedback, can improve most metrics, including AI feedback, compared to pre-trained models and supervised finetuning (SFT) baseline that does not leverage any reward signal. In addition, RWR and DPO exhibit better generalization to hold-out prompts than SFT. These can justify and encourage RL-finetuning with feedback to improve the qualities we are interested in for text-to-video generations. Comparing RWR and DPO, DPO increases all the evaluation metrics and often achieves better performance than RWR (7 in 10 metrics), which implies that subtle algorithmic choices, such as a direction of KL-regularization, on the unified objective would be a significant difference for RL-finetuning.

## 4.3 VLMs Enhance Dynamic Scene Quality

We here aim to identify what kind of algorithms and metrics can improve the perceptual quality of dynamic object interactions in text-to-video models. Table 2 provides the evaluation among all the combinations of algorithms and rewards, where the correctness and quality of dynamic scenes are assessed through binary feedback by VLMs and humans and also the representative quantitative video evaluation from VBench (Huang

et al., 2024). The results reveal that, while all the rewards could realize the improvement, RL-finetuning with AI feedback from Gemini (RWR-AIF and DPO-AIF) achieve the best performance, compared to any metric-based single rewards; RWR-AIF achieves +3.8% in Gemini evaluation (52.66% → 56.41%), +2.4% in GPT evaluation (44.45% → 46.80%), and +11.8% in human evaluation (19.38% → 31.09%). DPO-AIF achieves +3.5%, +1.8%, and +14.7% as an absolute gain. Furthermore, AI feedback from Gemini also improves four VBench scores the best, focusing on consistency and smoothness. The results highlight that AI feedback could show a great generalization across the metrics and that VLMs work as a proxy of labor-intensive human raters to improve the quality of dynamic object interactions (Figure 4).

Comparing RWR and DPO, RWR tends to achieve better performance on the train split while also exhibiting the *over-fitting* behaviors, where the performance against the test splits degrades from the pre-trained models. On the other hand, DPO does not face over-fitting and robustly aligns the output to be preferable, ensuring the generalization. Because SFT faces over-fitting issues too, we guess the lack of negative gradients that push down the likelihood of bad samples might cause insufficient generalization (Tajwar et al., 2024).

**Iterative RL-Finetuning.** Our recipe (Figure 2) can be extended to iterative finetuning, which can resolve distribution shift issues in offline finetuning (Matsushima et al., 2021; Xu et al., 2023b) and leads to further alignment to preferable outputs. Figure 5 (left) shows that RWR-AIF continually improves its output from self-generations (52.66% → 56.41% → 58.83%), while DPO-AIF is saturated (52.66% → 56.17% → 56.25%). This can be because the paired data from one-iteration DPO becomes equally good due to the overall improvement; it is hard to assign correct binary preferences.

**Comparison to Open Models.** Our preliminary experiments imply that even SoTA open models are not enough to generate good dynamics (Figure 1). VideoCrafter (Chen et al., 2024) and VADER (Prabhudesai et al., 2024) could not generate movements, such as *taking something*, and *burying*. We also provide both quantitative (Figure 5; right) and qualitative evaluations (Appendix F) of CogVideoX (Yang et al., 2024c) and Wan 2.1 (Team Wan et al., 2025). While generated videos are photorealistic and have high visual quality, they are often static and could not reflect the description of object movements in the prompts. The AI feedback results quantify the insufficiency of recent open models. Due to the lack of prior ability to express the dynamic scenes, a signal of improvements with open models is not observed yet, which motivated us to prepare a model that can simulate realistic movements from scratch.

**Analysis on Generated Object Movement.** We analyze the trend of generated video before/after RL-finetuning among five categories of challenging object movement. Figure 6 (left) shows AI preference (above) and absolute improvement (below) from pre-trained models per category. Generally, text-to-video models generate preferable videos of deformable objects (DO) and directional movement (DM), which are often complete with two-dimensional movement from the first frame. In contrast, they may not be good at modeling multi-step interactions, the appearance of new objects, and spatial three-dimensional relationships, which often occur in object removal (OR), multiple objects (MO), and falling down (FD). For instance, Figure 6 (right) requires multi-step interactions, such as opening a drawer, picking up a bottle opener, and putting it in the drawer, but the generation is stuck in the first step (see Appendix J for other failures). RL-finetuning notably improves video generation in the category of multiple objects and falling down, where it is relatively easy for VLMs to judge if the video scene is correct or not.

## 4.4 Connection to Human Evaluation

We first analyze the correlation between human preference and automated feedback, such as AI feedback from VLMs, and metric-based rewards (CLIP, HPSv2, PickScore, and Optical Flow). As done in Table 2 for AI/human evaluation, we first measure the average of each metric per algorithm-feedback combination (such as SFT, RWR-CLIP, DPO-AIF, etc), and compute Pearson correlation coefficient to the human preference (Figure 7; left). The results reveal that the performance measured with AI preference from VLMs has the most significant positive correlation to the one with human preference ($R = 0.746$; statistically significant with $p \leq 0.01$), while others only exhibit weak correlations, which supports the observation that optimizing AI feedback works as a proxy for supervised signals from humans.

Moreover, we find that finetuning metrics-based rewards with DPO, such as HPSv2 or PickScore, faces over-optimization (Azar et al., 2023; Furuta et al., 2024), where the metrics are improving while the generated

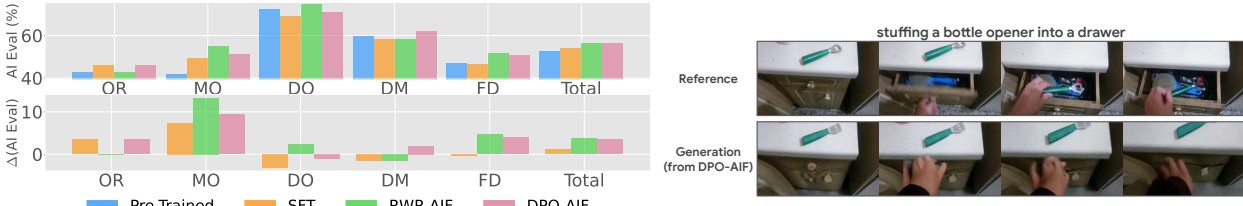

Figure 6: (**Left**) We analyze preferable video generation per category with AI evaluation (above) and absolute improvement from the pre-trained models (below). In general, text-to-video models can generate high-quality videos of deformable object and directional movement, while they are not good at generating the scene of object removal, multiple objects, and falling down. RL-finetuning with AI feedback significantly increases preferable outputs in multiple objects, and falling down categories. (**Right**) Example of failure generation after RL-finetuning, which could not represent multi-step interaction. See Appendix J for other failure cases.

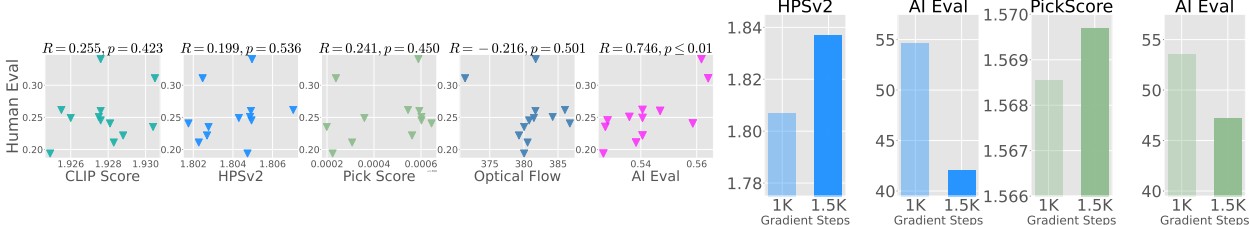

Figure 7: (**Left**) Pearson correlation coefficient between the performance with human preference and with other automated feedback, among 12 algorithm-reward combinations. AI preference from VLMs has the best positive correlation to human preference ($R = 0.746$; statistically significant with $p \leq 0.01$). (**Right**) Over-optimization issues in metric-based feedback. While DPO-HPSv2 could improve HPSv2 as the gradient step increases, the ratio of video accepted by VLMs significantly decreases. See Appendix I for the examples.

videos are visually worsened with incorrect events. For instance, in Figure 7 (right), while DPO-HPSv2 could improve HPSv2 as the gradient step increases, the ratio of acceptable video in the AI evaluation significantly decreases, which implies that metric-based rewards, even if representing human preferences, may not always be suitable for the correctness of events in video. It is essential for scalable text-to-video improvement to incorporate the supervision process by the capable VLMs, not limited to leveraging automated external feedback just for finetuning.

## 4.5 Takeaways for Algorithm Choices

Compared to feedback choices, where AI feedback is the best among others, algorithm choices have both positive and negative aspects. We here recap takeaways and recommendations of algorithmic designs for practitioners.

**Forward-EM-Projection.** Iterative RWR can continually improve the AI preference, while reverse-BT-projection gets saturated. This is promising to further scale up our methods. Even in offline finetuning, RWR is good at optimizing a metric within a specific set of prompts used for training, which is beneficial in practical use cases. In contrast, this also means that RWR exhibits overfitting in offline finetuning. In case you need to consider generalization, we may not recommend using it without online samples.

**Reverse-BT-Projection.** DPO improves most of the metrics better than forward-EM-projection because there is a negative gradient in the derivative of Equation 7 such as $\nabla_\theta \mathcal{J}_{\text{r-BT-neg}} \propto \beta \mathbb{E}[\nabla_\theta \log p_\theta(\mathbf{x}_0^{(2)} \mid \mathbf{c})]$, where $\mathbf{x}_0^{(1)} \succ \mathbf{x}_0^{(2)}$, which effectively pushes down the likelihood of undesirable generations. However, DPO with metric-based reward often falls into over-optimization, where the metrics improve while the generated videos are visually worsened. The performance of iterative finetuning also gets saturated soon, probably because the paired data after one iteration is equally good due to the overall improvement. It is noisy to assign binary preferences correctly.

**Summary.** In case you are only allowed to access an offline dataset, or in case you are interested in optimizing metric-based feedback, reverse-BT-projection can be the first choice. In case you have a budget to iterate the process, or in case you are interested in optimizing metrics within a specific set of prompts (less considering the generalization), it is worth employing forward-EM-projection.

## 5 Discussion and Limitation

Prior works have actively leveraged direct reward gradient from the differentiable metric-based rewards to align text-to-image (Prabhudesai et al., 2023; Clark et al., 2024), or even text-to-video models (Prabhudesai et al., 2024). In contrast, a recent study in LLMs (Zhang et al., 2024a) has argued that generative reward modeling can benefit from several advantages of LLMs and achieves better performance. The comprehensive analysis of AI feedback from the generative VLMs, compared to differentiable rewards is an important topic.

Due to the cost of querying VLMs online for AI feedback, the lack of a standard recipe to train a text-video reward, and the unstable behavior of policy gradient methods, this paper focuses on offline and iterative RL-finetuning. Considering the performance gain coming from online samples, it is a natural yet important direction to resolve the bottlenecks above and extend ours to naive online RL methods (Fan et al., 2023; Black et al., 2024). While we observed the insufficiency of the current SoTA open models due to the lack of prior knowledge on the dynamic scenes, other SoTA product models, such as Sora (OpenAI, 2024) or Veo (Google DeepMind, 2025), can generate seemingly fine video, yet still persist famous dynamics failures, such as unnaturally emerging a plastic chair from the desert. **As is model- and architecture-agnostic, we believe that our post-training can be widely applicable and beneficial to any capable models that still face failure modes.**

## 6 Related Works

**RL for Text-to-Image Generation.** Inspired by RL with human feedback for LLMs (Ouyang et al., 2022), there has been a great interest in employing RL-finetuning to better align text-to-image diffusion models (Lee et al., 2023b; Kim et al., 2024b). RL-finetuning for diffusion models has a lot of variants such as policy gradient (Fan et al., 2023; Uehara et al., 2024; Black et al., 2024; Zhang et al., 2024c;b) based on PPO (Schulman et al., 2017), offline methods (Wallace et al., 2023; Lee et al., 2023b; Liang et al., 2024; Yang et al., 2023a; Na et al., 2024; Yuan et al., 2024; Dong et al., 2023) based on DPO (Rafailov et al., 2023) or RWR (Peters & Schaal, 2007; Peng et al., 2019), and direct reward gradients (Clark et al., 2024; Prabhudesai et al., 2023). These work often optimize compressibility (Black et al., 2024), aesthetic quality (Ke et al., 2023), or human preference (Xu et al., 2023a; Li et al., 2024a), and show that RL-finetuning effectively aligns pre-trained diffusion models to specific downstream objectives. Our work focuses on aligning text-to-video models, which can be more challenging as videos have much complex temporal information, and it is unclear whether feedback developed for aligning text-to-image models can be directly applied to text-to-video.

**RL for Text-to-Video Generation.** There have also been a few recent works that explored finetuning text-to-video models with reward feedback (Prabhudesai et al., 2024; Li et al., 2024b; Yuan et al., 2023). These works leverage open text-to-video models (Wang et al., 2023; Blattmann et al., 2023a; Zheng et al., 2024b; Chen et al., 2024), and off-the-shelf text-to-image reward models, whose gradients are used to align the models to aesthetic or visual quality objectives rather than the dynamics in the scenes. However, because there is no standard protocol to collect comprehensive feedback for reward modeling, the reward to directly evaluate generated video has rarely been considered. Furthermore, policy-gradient-based methods are generally not stable (Rafailov et al., 2023). Our work differs from existing work in exploring a wide array of feedback to see if they can optimize the object movement rather than visual style, leveraging offline learning to bypass the need for learning a reward model, as well as considering AI feedback from VLMs as a proxy for human preference to generated videos, which has great potential as VLMs and video generation continue to improve.

Especially, there are several concurrent works that has applied DPO to text-to-video generation (Liu et al., 2024a; Wu et al., 2025). VideoDPO (Liu et al., 2024a) leverages standard DPO algorithm while proposing how we put preference labels from VBench scores (Huang et al., 2024); the evaluation is also from VBench.

In contrast, our paper first sheds the lights on the algorithmic equivalence between DPO and RWR, and comparing their performances, when trained with AI feedback; the evaluation is based on AI and human feedback, and VBench. When using VBench as a reward, improving VBench is relatively obvious. However, we demonstrate that AI feedback can improve human feedback and VBench. DenseDPO (Wu et al., 2025) proposes to leverage per-frame dense preference scores, which might be provided by VLMs. While DenseDPO considers overall preference, our paper sheds the light on dynamic object interaction, which persists even in SoTA production models and is challenging to be resolved by simply scaling data and models.

**AI Feedback for LLMs.** A wide set of work has explored leveraging AI feedback generated by LLMs to further improve or align generations of the same or a different LLM (Zheng et al., 2024a; Bai et al., 2022; Kim et al., 2023; Ling et al., 2024; Agarwal et al., 2024). There has also been recent work on extracting reward information from VLMs (Venuto et al., 2024; Rocamonde et al., 2023; Yan et al., 2024). Different from them, we explore the ability of long-context VLMs, such as Gemini-1.5 (Gemini Team, 2023) or GPT-4o (OpenAI, 2023), to provide feedback on various aspects of generated videos such as physical plausibility, consistency, and instruction following. While the scalable qualitative evaluation of video generation has been a long-standing problem (He et al., 2024; Liu et al., 2023; Wu et al., 2024a; Liao et al., 2024; Dai et al., 2024; Miao et al., 2024), we demonstrate that VLMs can be an automated solution, and AI feedback can provide informative signals to improve the dynamic scene generation.

**Modeling Object Interaction in Video Generation.** Many papers have studied how to model the accurate object interaction or spatial consistency in video generation so far (Sun et al., 2020): for example, by incorporating optical flow (Chefer et al., 2025) or key point prediction (Jeong et al., 2024), and modifying guidance methods in diffusion models (Hyung et al., 2024). Such recent papers rely on the evaluation with some automated metrics, such as VBench, while we employ both VLMs and human evaluation. Moreover, one of our important contribution is to show that RL-finetuning with VLMs can improve the contents of video beyond visual style, typically studied in prior RL works. We believe that this is an orthogonal contribution to the papers above.

## 7 Conclusion

We thoroughly examine design choices to improve the dynamic scenes in text-to-video generation. Our proposal, (iterative) RL-finetuning with AI feedback from VLMs, enhances the quality of dynamic scenes best, rather than other metric-based rewards. RL-finetuning may mitigate the issues in modeling multi-step interactions, the appearance of new objects, and spatial relationships. We hope this work helps further text-to-video generation for dynamic scenes.

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

# Appendix

## A    Details of Model Training

For the experiments, we first pre-train a video diffusion model (He et al., 2022), based on 3D-UNet (Özgün Çiçek et al., 2016), with Something-Something-V2 (Goyal et al., 2017; Mahdisoltani et al., 2018), which has **160K prompt-video pairs** in the train split and 8.5K pairs in the validation split. As explained in Section 4.1, we prepare 5×24 prompts for training and 5×8 prompts for evaluation from the original validation split. Then, we generate 128 samples per each prompt from pre-trained models to collect the data for finetuning ($5 \times 24 \times 128 = 15360$ samples in total). We follow He et al. (2022) for the base architecture and training setup. Specifically, the architecture is 3D-UNet with 3 residual blocks of 512 base channels and channel multiplier [1, 2, 4], attention resolutions [6, 12, 24], attention head dimension 64, and conditioning embedding dimension 1024. To support first-frame conditioning, we replicate the first-frame across all future frame indices and concatenate the replicated first-frame channel-wise to the noisy data. Following Yang et al. (2024a), we train a base (**1.6B parameters**; 16×24×40) and super-resolution model (**1.4B parameters**; 16×24×40 → 16×192×320), conditioned with embeddings from T5-XXL (Raffel et al., 2020) (sequence length = 64, embedding dim = 4096), and only finetune a base model with RL. We use Adam (Kingma & Ba, 2014) with linear scheduling (learning rate = $1e - 4$, gradient clipping = 1.0, warm-up steps = 10000). The batch size is 64. The gradient steps for pre-training are 125K, and those for RL-finetuning are 10K. RL-finetuning additionally requires linear reward transform, $r_{\text{lin}}(\mathbf{x}_0, \mathbf{c}) = \eta r(\mathbf{x}_0, \mathbf{c}) + \gamma$, as described in Section 3.2. Following Wallace et al. (2023), DPO uses $\beta = 5000$. Other parameters are shared between pre-training and RL-finetuning. We save a checkpoint per 500 steps and report the best results among the last 5 checkpoints. We have used cloud TPU-v3, which has a 32 GiB HBM memory space with 128 cores. The pre-training takes about 4–5 days, and RL-finetuning takes half a day.

## B    Example of Something-Something-V2

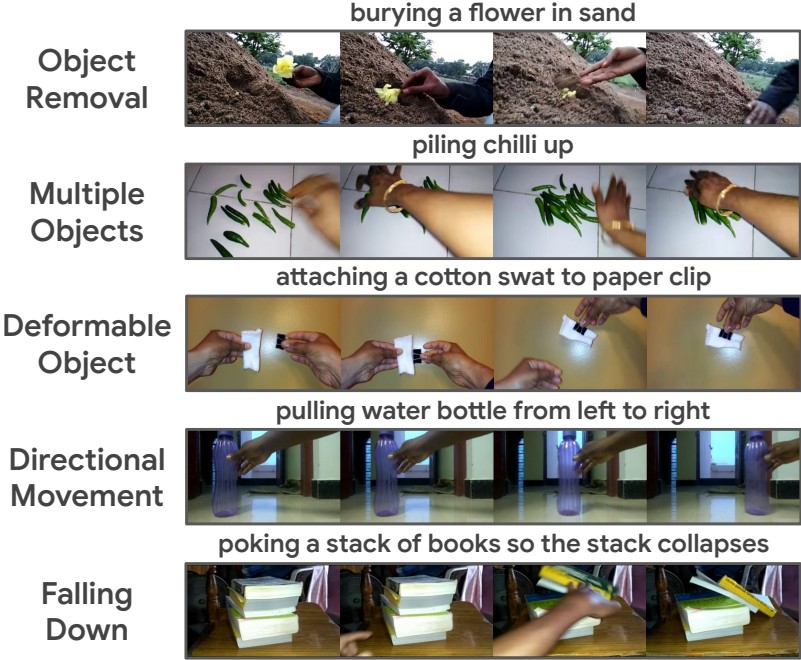

Figure 8: Additional example videos from Something-Something-V2. We focus five principles of challenging object movements: object removal (OR), multiple objects (MO), deformable object (DO), directional movement (DM), and falling down (FD).

## C  Prompts for Challenging Object Movements

### Object Removal (Train)

1. digging key out of sand
2. opening a drawer
3. putting coca cola bottle onto johnsons baby oil bottle so it falls down
4. removing beetroot , revealing cauliflower piece behind
5. uncovering pencil
6. wiping foam soap off of cutting board
7. burying flower in leaves
8. closing a bowl
9. plugging airwick scented oil diffuser into plugging outlet but pulling it right out as you remove your hand
10. burying a flower in sand
11. digging a leaf out of sand
12. showing that clip box is empty
13. pulling crucifix from behind of vr box
14. stuffing a ticket into a wooden box
15. taking seasor out of tin
16. glassess falling like a rock
17. rolling pen on a flat surface
18. uncovering a key
19. stuffing key into cup
20. removing red bulb , revealing blue marble behind
21. digging remote control out of sand
22. taking a pen out of the book
23. tilting wooden box with car key on it until it falls off
24. burying tomato in blanket

### Object Removal (Test)

1. taking cellphone out of white bowl
2. taking rose bud from bush
3. taking paper out of cigarette can
4. closing box
5. stuffing a bottle opener into a drawer
6. taking gas lighter out of cigarette can
7. scooping banana juice up with spoon
8. taking one of many coins

### Multiple Objects (Train)

1. lifting phone with pen on it
2. putting six markers onto a plate
3. putting cellphone , usb flashdisk and gas lighter on the table
4. putting 3 pencil onto towel
5. putting 4 blocks onto styrofoam sheet
6. moving cup and tin closer to each other
7. pushing calculator with marker pen
8. moving a candle and another candle away from each other
9. putting 4 pencils onto blanket
10. moving cup and fork away from each other
11. pushing avacado with book
12. putting a box , a pencil and a key chain on the table
13. moving a glass and a glass closer to each other
14. stacking three legos
15. moving mouse and brush away from each other
16. putting marker pen on the edge of plastic water cup so it is not supported and falls down
17. putting 4 pens onto a paper

18. moving plastic box and plastic box so they pass each other
19. stacking 4 numbers of cassette
20. pretending to close water tap without actually closing it
21. failing to put a drumstick into a purse because a drumstick does not fit
22. putting three shot glasses onto a box
23. taking one body spray of many similar
24. piling chilli up

### Multiple Objects (Test)

1. moving coin and napkin away from each other
2. moving lego away from mouse
3. putting lighter into shoe
4. putting spoon and flower on the table
5. attaching lid to sketch pen
6. putting cello tape onto powder container so it falls down
7. moving tv tuner and orange closer to each other
8. putting coins into bowl

### Deformable Object (Train)

1. folding cloth
2. folding a rag
3. tearing paper into two pieces
4. twisting ( wringing ) shirt wet until water comes out
5. tearing receipt into two pieces
6. moving adhesive tape down
7. folding a short
8. unfolding purse
9. tearing paper into two pieces
10. squeezing paper
11. folding paper
12. unfolding cloth
13. tearing tissues into two pieces
14. ziplock bag falling like a feather or paper
15. tearing a piece of paper into two pieces
16. unfolding floor mat
17. unfolding newspaper
18. squeezing toothpaste
19. tearing a leaf into two pieces
20. tearing paper just a little bit
21. spreading leaves onto floor
22. folding letter
23. squeezing a nylon bag

### Deformable Objects (Test)

1. tearing paper into two pieces
2. unfolding dish towel
3. unfolding a piece of paper
4. covering glue stick with tissue
5. unfolding blouse
6. stuffing a sock into a jar
7. attaching a cotton swat to paper clip
8. stacking three dish rags
9. folding winter cap

**Directional Movement (Train)**

1. pulling charger from right to left
2. pushing glove from left to right
3. pulling a tissue box from right to left
4. moving banana away from the camera
5. pushing cup from left to right
6. turning a calculator upside down
7. laying spray bottle on the table on its side , not upright
8. turning makeup product upside down
9. pulling trigonal clip from left to right
10. pushing newspaper from right to left
11. pulling scissors from right to left
12. putting a glass upright on the table
13. pushing glasses from right to left
14. pushing calculator from left to right
15. pushing brush from left to right
16. moving pencil across a surface without it falling down
17. pulling fork from right to left
18. pushing brush from left to right
19. pulling water bottle from left to right
20. pulling fabric from left to right
21. pulling ticket from left to right
22. turning container upside down
23. laying beer bottle on the table on its side , not upright
24. dropping wallet behind flower vase

**Directional Movement (Test)**

1. pulling a paper bag from right to left
2. pushing rubik ' s cube from right to left
3. pulling a mug from right to left
4. moving cellphone and fork so they pass each other
5. pushing screwdriver from right to left
6. pretending to close a book without actually closing it
7. dropping a battery in front of a teddy bear
8. pushing calculator from left to right

**Falling Down (Train)**

1. putting marker pen onto plastic bottle so it falls down
2. putting spoon on the edge of glass bottle so it is not supported and falls down
3. pushing a bottle so that it falls off the table
4. pushing key onto basket
5. pushing pocket notebook so that it falls off the table
6. pushing a flashlight so that it falls off the table
7. dropping dumbbell in front of basket
8. dropping toy duck next to teddy bear
9. big book falling like a rock
10. pulling plastic bowl onto floor
11. picking a watch up
12. throwing flask
13. tilting box with tube on it until it falls off
14. putting fork onto candle so it falls down
15. putting foil casserole dish upright on the table , so it falls on its side
16. dropping a ball in front of a book
17. tilting wooden scale with scissor on it until it falls off
18. tipping spray paint can with cap over , so cap falls out
19. moving belt across a surface until it falls down

20. poking an apple so that it falls over
21. lifting yellow marker up completely , then letting it drop down
22. poking menu card so that it falls over
23. tilting tray with buscuit on it until it falls off
24. dropping toothbrush onto stool

## Falling Down (Test)

1. pouring beer into glass
2. spilling water next to a plastic bottle
3. lifting a surface with a pawn on it until it starts sliding down
4. tilting smartphone backcover with cough syrup on it until it falls off
5. pushing digital multimeter so that it falls off the table
6. pushing a purse off of an ottoman
7. letting spray paint can roll down a slanted surface
8. poking a stack of books so the stack collapses

## D    Prompt for AI Feedback from VLMs

Task: You are a video reviewer evaluating a sequence of actions presented as eight consecutive images in the video below. You are going to accept the video if it completes the task and the video is consistent without glitches.

Inputs Provided:

Textual Prompt: Describes the task the video should accomplish.

Sequence of Images (8 Frames): Represents consecutive moments in the video to be evaluated.

Evaluation Process:

View and Analyze Each Frame: Examine each of the eight images in sequence to understand the progression and continuity of actions.

Assess Overall Coherence: Consider the sequence as a continuous scene to determine if the actions smoothly transition from one image to the next, maintaining logical progression.

Check for Physical Accuracy: Ensure each frame adheres to the laws of physics, looking for any discrepancies in movement or positioning.

Verify Task Completion: Check if the sequence collectively accomplishes the task described in the textual prompt.

Identify Inconsistencies: Look for inconsistencies in object movement or overlaps that do not match the fixed scene elements shown in the first frame.

Evaluation Criteria:

Accept the sequence if it is as a coherent video which completes the task.

Reject the sequence if any frame fails to meet the criteria, showing inconsistencies or not achieving the task. Reject even if there are the slightest errors. Do not be too strict in accepting the videos.

Response Requirement:

Provide a single-word answer: Accept or Reject.

Textual Prompt: {instruction}

Video: {video_tokens}

# E    Pseudo Algorithm for RL-Finetuning from Feedback

---

**Algorithm 1** Offline RL-Finetuning Text-to-Video Models with Feedback

---

**Input:** text-to-video model $p_\theta$, dataset for pre-training $D_{\text{pre}} = \{\mathbf{x}_0^{(i)}, \mathbf{c}^i\}_{i=1}^N$, a set of conditional features for
   finetuning $C_{\text{fine}} = \{\mathbf{c}^j\}_{j=1}^M$, reward model $r(\mathbf{x}_0, \mathbf{c})$

1: Pre-train text-to-video model $p_{\text{pre}}(\mathbf{x}_0 \mid \mathbf{c})$ with $D_{\text{pre}}$ by minimizing $\mathcal{J}_{\text{DDPM}}$
2: Generate video $\hat{\mathbf{x}}_0^j \sim p_{\text{pre}}(\cdot \mid \mathbf{c}^j)$ conditioned on $\mathbf{c}^j \in C_{\text{fine}}$ to collect dataset for finetuning $D_{\text{fine}}$
3: Label AI feedback $r(\hat{\mathbf{x}}_0^j, \mathbf{c}^j)$ to $D_{\text{fine}}$ with VLMs (or metric-based reward such as CLIP)
4: (for RWR) Minimize $\mathcal{J}_{\text{f-EM}}$ (Equation 6) with $D_{\text{fine}}$
5: (for DPO) Minimize $\mathcal{J}_{\text{r-BT}}$ (Equation 7) with a paired dataset from $D_{\text{fine}}$

---

# F    Failure Mode of SoTA Open Text-to-Video Models

Our preliminary experiments demonstrate that even SoTA open models, such as VideoCrafter (Chen et al., 2024), CogVideoX (Yang et al., 2024c), and Wan 2.1 (Team Wan et al., 2025), are still not enough to generate good dynamics. For instance, Figure 1 and Figure 9 show examples of dynamic scenes generated by VideoCrafter (Chen et al., 2024) or VADER (Prabhudesai et al., 2024). We observe that they cannot generate movements, such as *falling off*, *tearing*, and *dropping*. Due to the lack of prior ability to generate the dynamic scenes, we could not observe the strong signals of the improvement from our preliminary experiments with open models.

In contrast, other SoTA product models, such as Sora (OpenAI, 2024), can generate seemingly good video, but they also have famous dynamics failures, such as unnaturally emerging a plastic chair from the desert. We believe our method can apply to any models if they can output dynamic scenes.

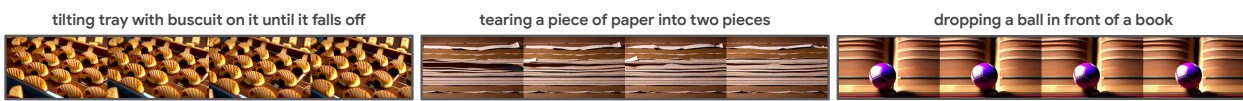

Figure 9: Generated videos from VideoCrafter (Chen et al., 2024), one of the SoTA open models. VideoCrafter fails to represent object movement in the prompts, such as *falling off* or *tearing*.

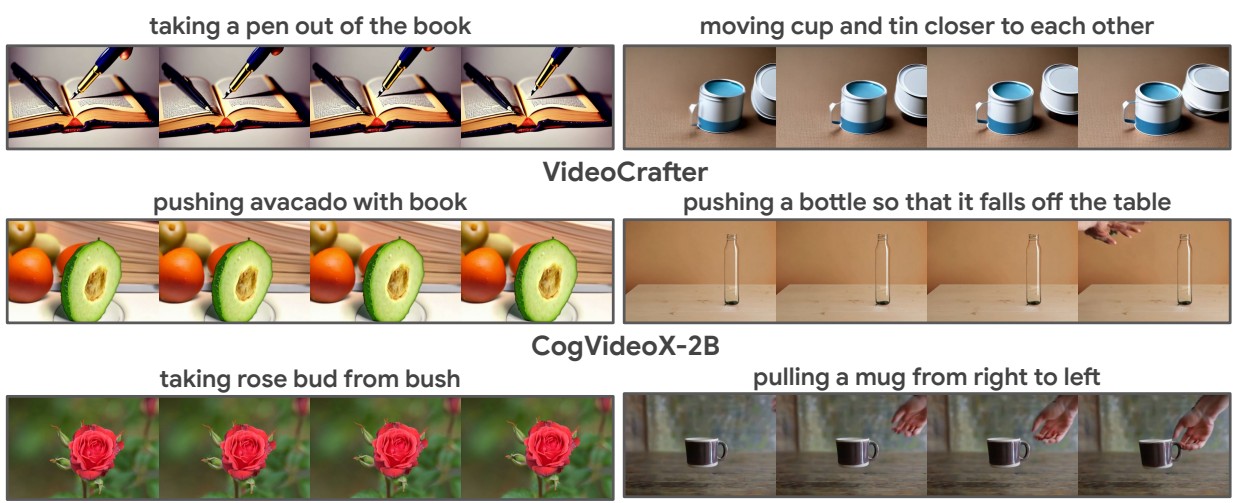

Figure 10: Examples from recent SoTA open models (VideoCrafter (Chen et al., 2024), CogVideoX-2B (Yang et al., 2024c), Wan 2.1-1.3B (Team Wan et al., 2025)). While the generated videos are photorealistic and have high visual quality, they are often static and could not reflect the description of the object movements in the prompts. See Figure 5 (right) for the quantitative evaluations.

## G    Performance Gap before and after RL-Finetuning

Table 3 shows FVD, FID, IS, and CLIP scores (summing image metric over frames) measured with a validation split of Something-Something-V2 dataset (about 8.5K prompts excluding $5 \times 32$ prompts used for RL-finetuning). As discussed in Lee et al. (2023b), conventional video metrics (FVD, FID, and IS) slightly drop, while text alignment assessed by CLIP is improved. Notably, leveraging reward signals is better than SFT overall. We believe that the original performance of pre-trained models is maintained within a reasonable range, and we successfully improve the capability to express the complex dynamic object interactions (Table 2), which is evaluated through AI feedback, human feedback, and scores in VBench (Huang et al., 2024).

|  | FVD ↓ | FID ↓ | IS ↑ | CLIP ↑ |
|---|---|---|---|---|
| **PT** | **10.78** | **4.27** | **3.58** | 0.2296 |
| **SFT** | 14.13 | 5.12 | 3.43 | 0.2298 |
| **RWR-AIF** | 13.63 | 4.96 | 3.47 | **0.2300** |
| **DPO-AIF** | 11.98 | 4.30 | 3.52 | 0.2299 |

Table 3: FVD, FID, IS, and CLIP score with Something-Something-V2-validation dataset (eval batch size: 4096).

## H    Example of Generated Video

In addition to Figure 4, Figure 11 provides examples of generated videos from pre-trained models, RWR-AIF, and DPO-AIF.

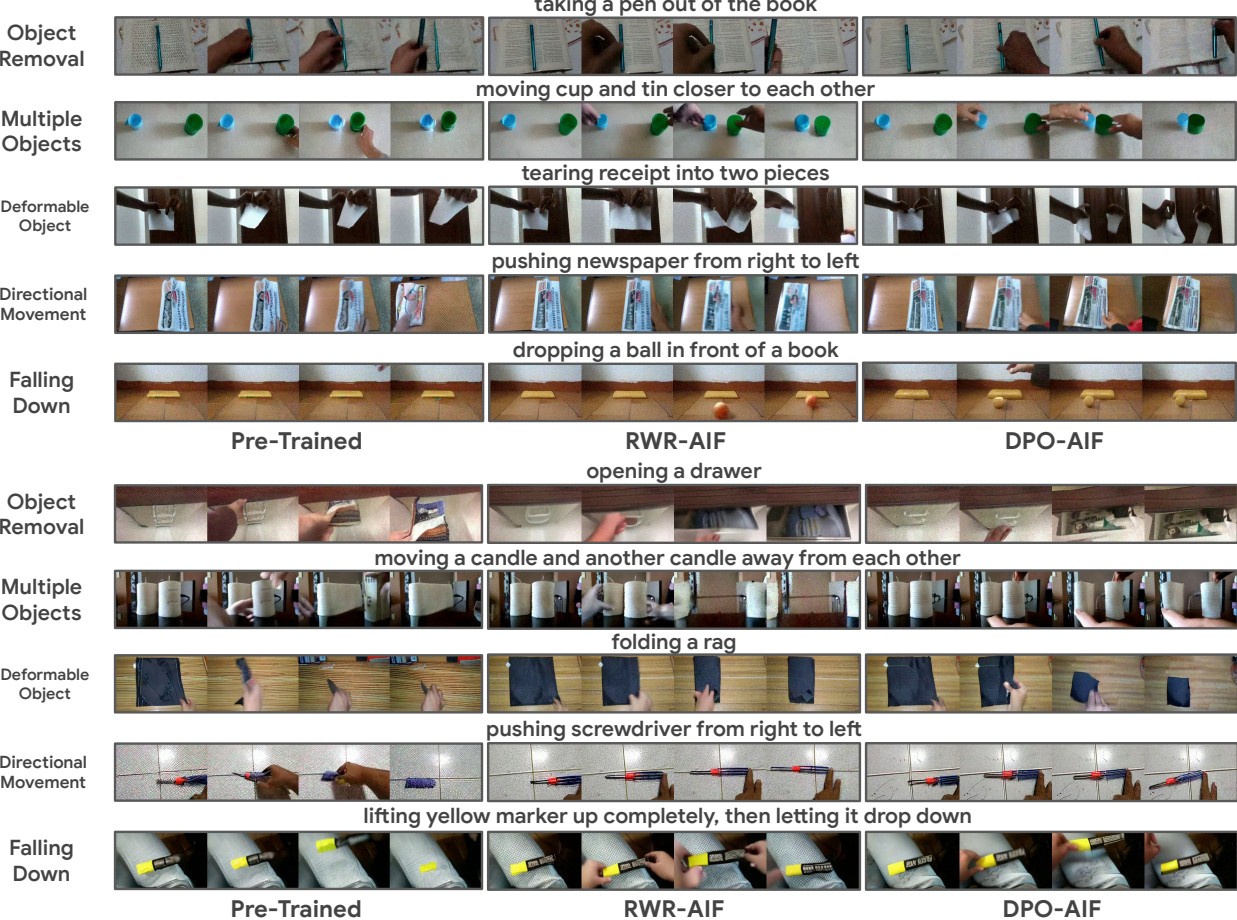

Figure 11: Generated videos from pre-trained models, RWR-AIF, and DPO-AIF.

## I  Failure Mode of Over-Optimization

As discussed in Section 4.3, we find that finetuning metrics-based rewards with DPO, such as HPSv2 or PickScore, faces over-optimization issues (Azar et al., 2023; Furuta et al., 2024), where the metrics are improving while the generated videos are visually worsened to the humans (Figure 12).

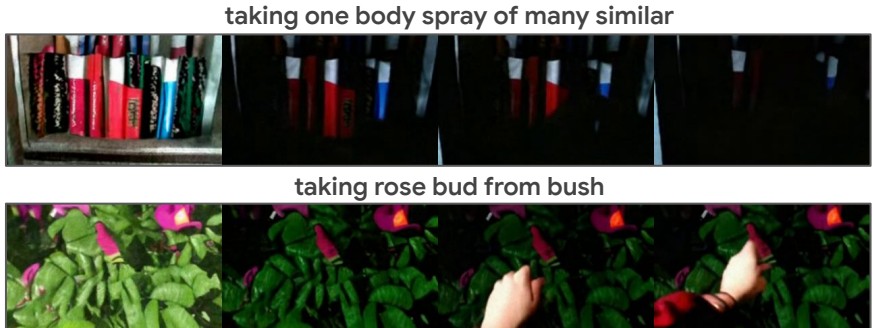

Figure 12: Generated videos from the models over-optimized with DPO-HPSv2. The model just makes the tone of the frames dark, without any improvement on the dynamics.

## J  Failure Mode of Dynamic Scene Generation

We show the failure generations after RL-finetuning in Figure 13, where the text-to-video models still suffer from modeling multi-step interactions (*stuffing a bottle opener into a drawer*), appearance of new objects (*putting 4 pencils onto blanket* and *unfolding purse*), and spatial three-dimensional relationship (*dropping wallet behind flower vase* and *tilting box with tube on it until it falls off*).

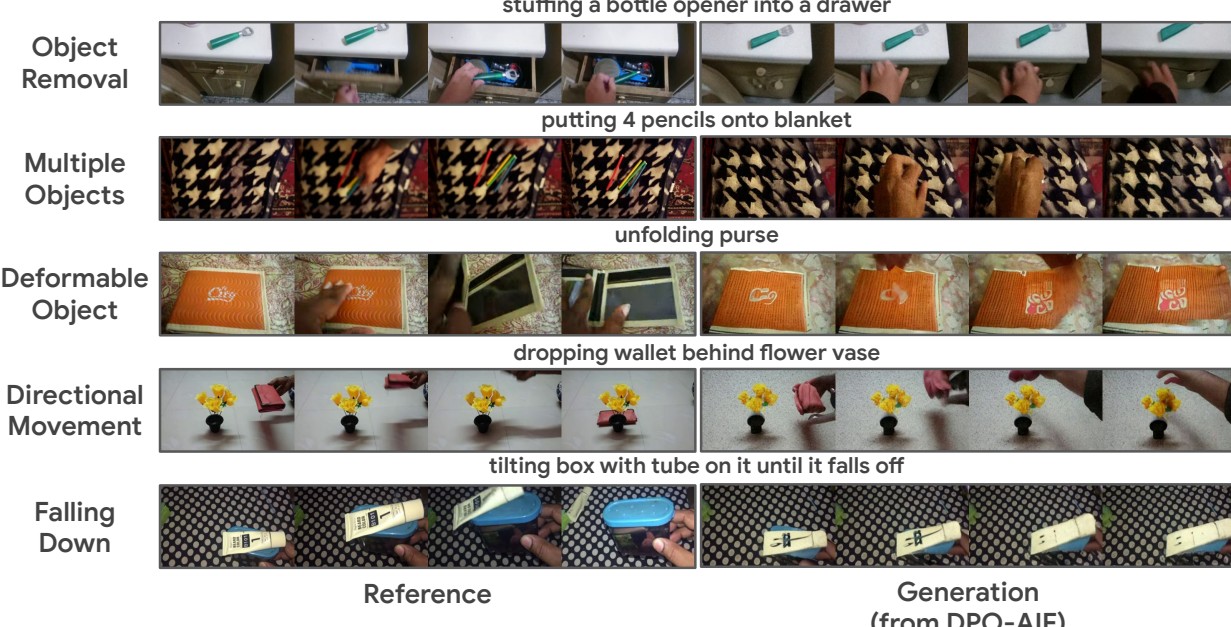

Figure 13: Failure generations after RL-finetuning. As presented above, the text-to-video models suffer from modeling multiple interactions, appearance of new objects, and spatial three-dimensional relationship.

# K    Extended Analysis of Correlation between Human Evaluation and Automated Feedback

In Section 4.4, we measure the average of each metric per algorithm-feedback combination (such as SFT, RWR-CLIP, DPO-AIF, etc), and compute Pearson correlation coefficient to the human preference (Figure 7; left). The results reveal that the performance measured with AI preference from VLMs has the most significant positive correlation to the one with human preference, which supports the observation that optimizing AI feedback can be the best proxy for optimizing human feedback.

In contrast, when we measure the correlation between the averaged human preference and automated feedback among the $32 \times 5 = 160$ prompts (Table 4), human preference exhibits weak positive correlation only to AI preference ($R = 0.231$; statistically significant with $p \leq 0.01$)), and others does not have notable correlations. This implies that even with the best choice – AI feedback from VLMs, the rationale of preference is still not perfectly aligned with human feedback (Wu et al., 2024b). This is because the evaluation of video is complex, and the multiple criteria are closely connected to each other, such as text-video alignment, aesthetics, appearance, colorization, smoothness, consistency, flickering, spatial relationship, temporal dependencies, etc (Huang et al., 2024). Humans can take all of them into consideration, but VLMs may only simulate a part of them. While we demonstrate the promising signal to leverage VLMs to improve the quality of dynamic object interaction in text-to-video models, we also call for the improvement of VLMs for quality judgments to be calibrated with human perception.

| Feedback | Correlation | $p$-value |
|---|---|---|
| **CLIP** | 0.011 | $p = 0.89$ |
| **HPSv2** | 0.163 | $p \leq 0.05$ |
| **PicScore** | 0.112 | $p = 0.16$ |
| **Optical Flow** | -0.107 | $p = 0.18$ |
| **AI Feedback** | **0.231** | $p \leq 0.01$ |

Table 4: Correlation between the averaged human preference and the averaged automated feedback, among $32 \times 5 = 160$ prompts. Even the best AI feedback only exhibits a weak positive correlation ($R = 0.231$ with $p \leq 0.01$). The rationale of preference from VLMs is not perfect to be aligned with humans, despite the promising quality improvement.

