# OpenReview forum: "Improving Dynamic Object Interactions in Text-to-Video Generation with AI Feedback"
_TMLR — Rejected by TMLR_

### Review · Reviewer_S4qK · 2026-02-02

**Summary Of Contributions:**

This paper investigates improving dynamic object interactions in text-to-video generation using external feedback. The technical contributions include:

* A unified probabilistic objective showing that RWR and DPO stem from the same framework, with the insight that there is no algorithmically dominant method—what matters is the property of reward and data.
* Using VLMs (Gemini-1.5-Pro) to provide binary AI feedback as reward signal for RL-finetuning.
* Defining five categories of challenging object movements and systematic comparison of algorithm-reward combinations.

Strengths:

* The unified framework provides conceptual clarity: the equivalence between RWR and DPO highlights that algorithm choice should depend on use case (RWR for iterative improvement, DPO for generalization), not assumed superiority.
* The finding that metric-based rewards (CLIP, HPSv2, PickScore) poorly correlate with human perception of dynamics, while VLM feedback is an important practical insight.
* The over-optimization phenomenon in DPO with metric-based rewards is a useful warning for practitioners.
* The analysis of failure modes (multi-step interactions, new object appearances, 3D spatial relationships) provides actionable guidance for future work.

Weaknesses:

* Experiments use a custom-trained 3D-UNet on Something-Something-V2, not mainstream models like CogVideoX or Wan.
* Binary feedback is coarse compared to recent dense preference methods.

**Audience:**

Yes

**Audience Explanation:**

The paper offers several insights valuable to the community: (1) the unified view of offline RL algorithms emphasizing reward/data properties over algorithm choice, (2) evidence that VLM feedback is a better proxy for human judgment than metric-based rewards for dynamic scenes, (3) characterization of when RWR vs. DPO should be preferred, and (4) identification of specific failure modes in current video generation. These findings are practically relevant as video generation models continue to improve.

**Broader Impact Concerns:**

No major concerns. A brief acknowledgment of potential misuse for generating misleading video content would be appropriate.

**Claims And Evidence:**

Yes

**Claims Explanation:**

Within the paper's experimental setup, the main claims are supported: the unified framework derivation is sound, AI feedback outperforms metric-based rewards, and the RWR vs. DPO trade-offs are empirically validated.
However, the scope of validation is limited. All experiments use a custom-trained 3D-UNet on Something-Something-V2 with 160 prompts. Figure 5 shows mainstream models (CogVideoX: 10.83%, Wan: 10.83%) perform poorly, but no experiment demonstrates the method can improve them. The claim that the approach is "model- and architecture-agnostic" requires validation on at least one widely-used open-source model. Additionally, comparison with concurrent methods like DenseDPO is discussed but not experimentally validated.

**Requested Changes:**

Major:

* Demonstrate the method on at least one mainstream open-source model (CogVideoX, Wan, etc.) If technical barriers prevent this, explain them clearly.
* Evaluate on additional prompts beyond Something-Something-V2 to validate generalization.
* Experimental comparison with dense feedback methods like DenseDPO, or ablation on feedback granularity (binary vs. segment-level).


Recommended:

* Include recent RL algorithms (e.g., GRPO) to strengthen the claim that algorithm choice is less important than reward/data properties.

---

### Review · Reviewer_wLNe · 2026-02-04

**Summary Of Contributions:**

This work investigates the use of RL finetuning with AI feedbacks to improve dynamic object interactions in T2V generation.

The authors make three contributions:

(1) they establish a unified probabilistic framework showing that offline RL algorithms (RWR and DPO) derive from the same objective;

(2) they propose using VLMs as perceptual evaluators to provide binary AI feedback on object dynamics;

(3) they conduct extensive experiments demonstrating that AI feedback outperforms traditional metric-based rewards like CLIP and HPS in improving realistic object interactions.

The work is evaluated on a curated dataset of 160 challenging prompts from Something-Something-V2 spanning five categories of object dynamics.

In summary, I believe it is a well-conducted work with solid theoretical foundation and extensive experiments.

**Audience:**

Yes

**Audience Explanation:**

T2V generation is a rapidly evolving field. The community is actively seeking methods to move beyond high-fidelity static scenes to dynamically consistent videos. RL with VLM feedbacks is a feasible solution for this challenge.

**Broader Impact Concerns:**

NO or VERY MINOR ethics concerns only.

**Claims And Evidence:**

Yes

**Claims Explanation:**

1. **Baselines**: The authors compare their models against pretrained and SFT baselines, and models like VideoCrafter and VADER.

2. **Evaluation**: The authors assess their method on human evaluation, VLM evaluation, and VBench, and the results generally support their claims.

3. **Ablation**: The comparison between different rewards clearly isolates the impact of the VLM-based feedback.

4. **Limitations**: Limitations have been discussed by the authors.

**Requested Changes:**

1. **Supplement results on advanced T2V backbones**: It would be highly valuable to verify if the proposed AI-feedback fine-tuning consistently improves recent large-scale T2V models like WAN 2.2-A14B.

2. **Clarification of "Equivalence"**: "RWR and DPO are equivalent" is a strong claim, and authors should elaborate on the clarification of it to avoid misleading readers.

3. **Discussion on Generalization**: The study is currently limited to the specific domain of Something-Something-V2. Please add a brief discussion on the feasibility of applying this pipeline to broader domains.

---

### Review · Reviewer_tT4S · 2026-02-13

**Summary Of Contributions:**

This paper proposes improving dynamic object interactions in text-to-video diffusion models through RL fine-tuning with external feedback, and makes two key contributions: first, it provides a unified probabilistic inference view showing that representative offline RL algorithms such as Reward-Weighted Regression (RWR) and Direct Preference Optimization (DPO) are derived from the same objective, clarifying that performance differences stem more from reward design than algorithmic form. second, it introduces binary AI feedback from vision-language models (e.g., Gemini) as a scalable proxy for human supervision, demonstrating that optimizing this feedback significantly improves realism, physical plausibility, and multi-object dynamic interactions compared to metric-based rewards like CLIP or optical flow. The work is strong in its theoretical unification, comprehensive evaluation (including human and VLM assessments), and focus on a persistent failure mode in video generation, but it depends on large proprietary VLMs, uses relatively coarse binary feedback, and is validated mainly on a 3B model, leaving open questions about scalability to larger production systems.

**Audience:**

Yes

**Audience Explanation:**

The paper addresses reinforcement learning fine-tuning for diffusion models, probabilistic interpretations of RL objectives, and alignment via automated feedback—all core topics within machine learning research. In particular, researchers working on generative modeling, diffusion models, multimodal learning, RL for generative models, and alignment methods would find the unified perspective on RWR and DPO valuable . The theoretical clarification that commonly used offline RL methods share a unified objective is broadly relevant beyond video generation, potentially informing work in text-to-image, LLM alignment, and other generative domains.

Additionally, the empirical study of AI feedback from vision-language models as a scalable alternative to human supervision connects to ongoing interest in AI feedback, self-improvement loops, and automated evaluation—areas of active discussion in the TMLR community. Even readers not focused specifically on video generation may find the insights about reward design, over-optimization of metrics, and the trade-offs between forward- and reverse-KL regularization informative.

While the application domain (text-to-video dynamics) is somewhat specialized, the methodological and theoretical contributions are general enough that at least a subset of TMLR’s audience would likely find the work relevant and informative.

**Broader Impact Concerns:**

The work improves realism and physical plausibility in text-to-video generation via RL fine-tuning with AI feedback . While technically motivated, increasing the realism of dynamic object interactions may lower the barrier for generating convincing synthetic videos, potentially enabling misinformation, fabricated evidence, or deceptive media. In addition, relying on large proprietary VLMs as automated “judges” raises concerns about bias amplification and lack of transparency, as their implicit assumptions may be encoded into the trained model. The broader impact discussion should more explicitly address misuse risks, evaluator bias, and potential safeguards.

**Claims And Evidence:**

Yes

**Claims Explanation:**

The submission provides clear and largely convincing evidence to support its claims. The core claim—that RL fine-tuning with AI feedback from vision-language models (VLMs) improves dynamic object interactions in text-to-video generation more effectively than metric-based rewards—is substantiated through comprehensive empirical evaluation . The authors compare multiple reward types (CLIP, HPSv2, PickScore, Optical Flow, and AI feedback), multiple algorithms (RWR and DPO), and include SFT baselines, train/test splits, iterative fine-tuning, and category-level analysis of challenging motion types. Improvements are consistently observed across VLM evaluation, human preference, and VBench metrics, which directly align with the paper’s stated objective of improving dynamic realism. The theoretical claim that RWR and DPO can be derived from a unified probabilistic objective is also clearly presented and supported by empirical observations regarding overfitting and generalization differences.

That said, some limitations slightly weaken the breadth of the evidence: results depend heavily on proprietary VLMs for both reward generation and evaluation, feedback is binary rather than fine-grained, and experiments are conducted on a 3B model rather than large-scale production systems. Thus, while the claims are well-supported within the presented experimental setting, further validation across larger models and more diverse setups would strengthen the generality of the conclusions.

**Requested Changes:**

While the paper addresses an important problem, my main concern is limited novelty. In its current form, I am not convinced that the methodological contribution is sufficiently new to justify acceptance.

### Critical Concerns

1. **Limited methodological novelty.**
   The core technical components—offline RL fine-tuning of diffusion models (e.g., RWR, DPO) and the use of AI feedback from large models—are already well-established in adjacent domains such as text-to-image and LLM alignment [1,2,3] . The paper largely adapts existing RL objectives and applies them to text-to-video generation. While the empirical study is thorough, the conceptual advance appears incremental rather than fundamentally new.

2. **Unified view of RWR and DPO is clarificatory but not groundbreaking.**
   The probabilistic unification of RWR and DPO is intellectually clean, but similar connections between KL-regularized RL objectives and preference optimization have been discussed in prior literature. It is unclear how much genuinely new theoretical insight is introduced beyond reformulation.

3. **AI feedback as reward is not a new paradigm.**
   Using strong foundation models to provide reward or preference signals has become a common approach in alignment research. The paper demonstrates that this works for dynamic video interactions, but this feels more like a domain transfer than a conceptual breakthrough.

4. **Empirical gains may reflect stronger evaluators rather than deeper modeling advances.**
   Since powerful VLMs are used both for reward and evaluation, the improvement could partially stem from better alignment to those specific evaluators, rather than representing a fundamentally new training principle.



### What Would Strengthen the Novelty Claim

* Clearer positioning against prior RL-for-diffusion and AI-feedback work, explicitly articulating what is fundamentally new rather than applied.
* Stronger theoretical insight beyond unification—e.g., new objective properties, convergence guarantees, or analysis specific to temporal diffusion.
* Demonstrating capabilities that were previously unattainable with existing methods, rather than incremental improvements.

[1] ImageReward: Learning and Evaluating Human Preferences for Text-to-Image Generation. NeurIPS 2023

[2] Unified Multimodal Chain-of-Thought Reward Model through Reinforcement Fine-Tuning. NeurIPS 2025

[3] VisionReward: Fine-Grained Multi-Dimensional Human Preference Learning for Image and Video Generation. AAAI 2026

---

### Decision · Action_Editor_NTzL · 2026-04-08

**Recommendation:** Reject

**Additional Comments:**

This paper investigates RL fine-tuning with AI feedback to improve dynamic object interactions in text-to-video generation. It offers a unified probabilistic view of RWR and DPO and proposes using VLM-based binary feedback as a reward signal. The topic is timely, and the findings are of interest (agreed by all reviewers).

However, two reviewers leaned towards rejecting this paper. I note that several reviewer concerns are narrow experimental scope, insufficient algorithmic coverage, and evaluation circularity.

I encourage the authors to resubmit after either (a) expanding experimental validation to at least one mainstream T2V model and additional RL algorithms, or (b) narrowing the claims to match the current experimental scope.

**Audience:**

Yes

**Audience Explanation:**

1. The unified probabilistic view connecting RWR and DPO offers conceptual clarity that is useful to researchers working on RL fine-tuning for generative models, not only in video but also in text-to-image and LLM alignment settings.
2. The empirical finding that popular metric-based rewards (CLIP, HPSv2, PickScore) correlate poorly with human perception of dynamic realism, while VLM-based binary feedback serves as a much better proxy, is a practical insight relevant to reward functions for generative model alignment.

**Claims And Evidence:**

No

**Claims Explanation:**

The core concern, shared across all three reviewers, is that all experiments are conducted on a single custom-trained 3B 3D-UNet model using only 160 prompts from Something-Something-V2. This narrow experimental scope is insufficient to support the paper's broader claims. For example, the feedback granularity is limited to binary labels, with no comparison against denser feedback methods such as DenseDPO (Reviewer S4qK).